# Experimental study platform for electrocatalysis of atomic-level controlled high-entropy alloy surfaces

Yoshihiro Chida [1] ✉, Takeru Tomimori[1], Tomoaki Ebata [1], Noboru Taguchi [2], Tsutomu Ioroi [2], Kenta Hayashi [1], Naoto Todoroki [1] & Toshimasa Wadayama [1]

High-entropy alloys (HEAs) have attracted considerable attention to improve performance of various electrocatalyst materials. A comprehensive understanding of the relationship between surface atomic-level structures and catalytic properties is essential to boost the development of novel catalysts. In this study, we propose an experimental study platform that enables the vacuum synthesis of atomic-level-controlled single-crystal high-entropy alloy surfaces and evaluates their catalytic properties. The platform provides essential information that is crucial for the microstructural fundamentals of electrocatalysis, i.e., the detailed relationship between multi-component alloy surface microstructures and their catalytic properties. Nanometre-thick epitaxially stacking layers of Pt and equi-atomic-ratio Cr-Mn-Fe-Co-Ni, the so-called Cantor alloy, were synthesised on low-index single-crystal Pt substrates (Pt/Cr-Mn-Fe-Co-Ni/Pt(hkl)) as a Pt-based single-crystal alloy surface model for oxygen reduction reaction (ORR) electrocatalysis. The usefulness of the platform was demonstrated by showing the outperforming oxygen reduction reaction properties of high-entropy alloy surfaces when compared to Pt-Co binary surfaces.

The atomic-level surface designs of various alloys are key for improving the surface catalytic properties that are essential for development of novel electrocatalytic materials because alloy surfaces are structurally more complicated from the viewpoints of kinds of constituent elements and their compositions. For example, Pt-based binary alloys have been intensively studied[1,2] because their catalytic properties determine the performance of polymer electrolyte membrane fuel cell (PEMFC) under harsh operating conditions such as strong acidity and severe potential fluctuations. Through these studies, broad consensus has been reached for high-performance catalyst surfaces relative to each of the various structural factors, e.g. constituent elements[3], alloy compositions[4], crystallinity order[5,6] and atomic arrangement[7,8]. However, even though remarkable developments have been achieved in atomic-resolution transmission electron microscopy[9,10], information for precise surface atomic structures even in Pt-based binary alloys remains insufficient because both the outermost- and near-surface of alloy catalysts are generally incredibly complicated even before use (in the pristine state). Furthermore, pristine and atomic-level surface structures, e.g. atomic arrangements and alloy compositions in the surface vicinities, dynamically change due to power generations[1,11]. Therefore, atomic- and nano-level explorations of alloy catalyst surfaces that are directly correlated with the activity and durability of catalysts have always been a challenging issue for material development in practical electrocatalysts.

High-entropy alloys (HEAs), which generally comprise more than five elements, have recently attracted considerable attention because of their unique thermodynamical[12] and chemical properties[13]. To date, various application studies have been reported for these HEA systems with respect to various types of electrocatalysis[14–21]. Meanwhile, only a

[1]Graduate School of Environmental Studies, Tohoku University, Sendai 980-8579, Japan. [2]National Institute of Advanced Industrial Science and Technology, Ikeda 563-8577, Japan. ✉e-mail: yoshihiro.chida.t8@dc.tohoku.ac.jp

few studies have been reported to clarify the catalytic and surface atomic-level structural relationships of HEA electrocatalysis probably because high-entropy factors increase so-called versatility, i.e., the structural and compositional complexities, such as atomic-level uniformity of elemental distribution and localised electronic properties. Therefore, considering the aforementioned current status of HEA studies, the surface properties of even more complex HEA atomic-level structures need to be understood, and precise surface tuning of the locations and compositions of constituent elements, which is particularly aimed at the electrocatalysis application fields, is indispensable. Although materials informatics[22–24] (MI) will be powerful for predicting the catalytic properties of such complex multi-component alloys, machine times, calculation conditions and, furthermore, reliable datasets for the calculation of HEA systems are still insufficient for practical novel catalyst development. Hence, systematic fundamental experimental studies based on atomic-level characterisations of HEA surfaces are essential. Thus, a combination of experimental studies and MI is indispensable to boost the development of high-performance HEA catalysts.

In this study, an experimental study platform for HEA electrocatalysis for the vacuum-synthesis of atomic-level controlled model catalyst surfaces is proposed in this study. Subsequently, the catalytic properties were evaluated and structural characterisations of the well-defined HEA surfaces were conducted (Fig. 1). Sample vacuum synthesis, which enables the fabrication of single-crystal HEA model catalyst surfaces using various combinations of constituent elements, alloy compositions and surface atomic arrangements, can be achieved by

sequential deposition of specific pure elements and/or alloys on metal single-crystal substrates using multiple arc-plasma-deposition (APD) sources installed in an ultra-high vacuum (UHV; <10⁻⁷ Pa) chamber. After the vacuum synthesis of the aimed model catalyst by controlling the surface atomic-level structures, the sample was transferred from the UHV chamber to an $N_2$-atmospheric electrochemical measurement environment without exposure to air using a home-built vacuum-transfer system[25]. This process was performed to minimise the influence of oxidation and/or contamination on the sample surfaces, particularly for the less noble elements, which dominate the essential electrocatalytic properties of the single-crystal surfaces of HEAs. The catalytic properties evaluated using the aforementioned experimental procedures would provide reliable training datasets for MI, e.g., various parameters including structural factors, kinds of constituent elements and compositions, annealing temperatures, etc.

This study focused on HEAs of Pt and an equi-atomic ratio HEA of Cr, Mn, Fe, Co and Ni with a face-centred cubic (FCC) crystal structure (the so-called Cantor alloy[26]) for PEMFC cathode-catalyst applications. The oxygen reduction reaction (ORR) properties (activity and structural stability) of the resulting APD of Pt/Cr-Mn-Fe-Co-Ni fcc low-index single-crystal surfaces synthesised on Pt single-crystal surfaces (Pt(hkl); hkl = 111, 110, 100) were investigated. Because the topmost surfaces of the resulting HEA are composed of low-index single-crystal domains of Pt with the same surface symmetries as those of the Pt single-crystal substrates, the electrocatalytic properties of the resulting well-defined HEA surfaces can be systematically evaluated and explained. The ORR properties of the Pt/Cr-Mn-Fe-Co-Ni/Pt(hkl) single-

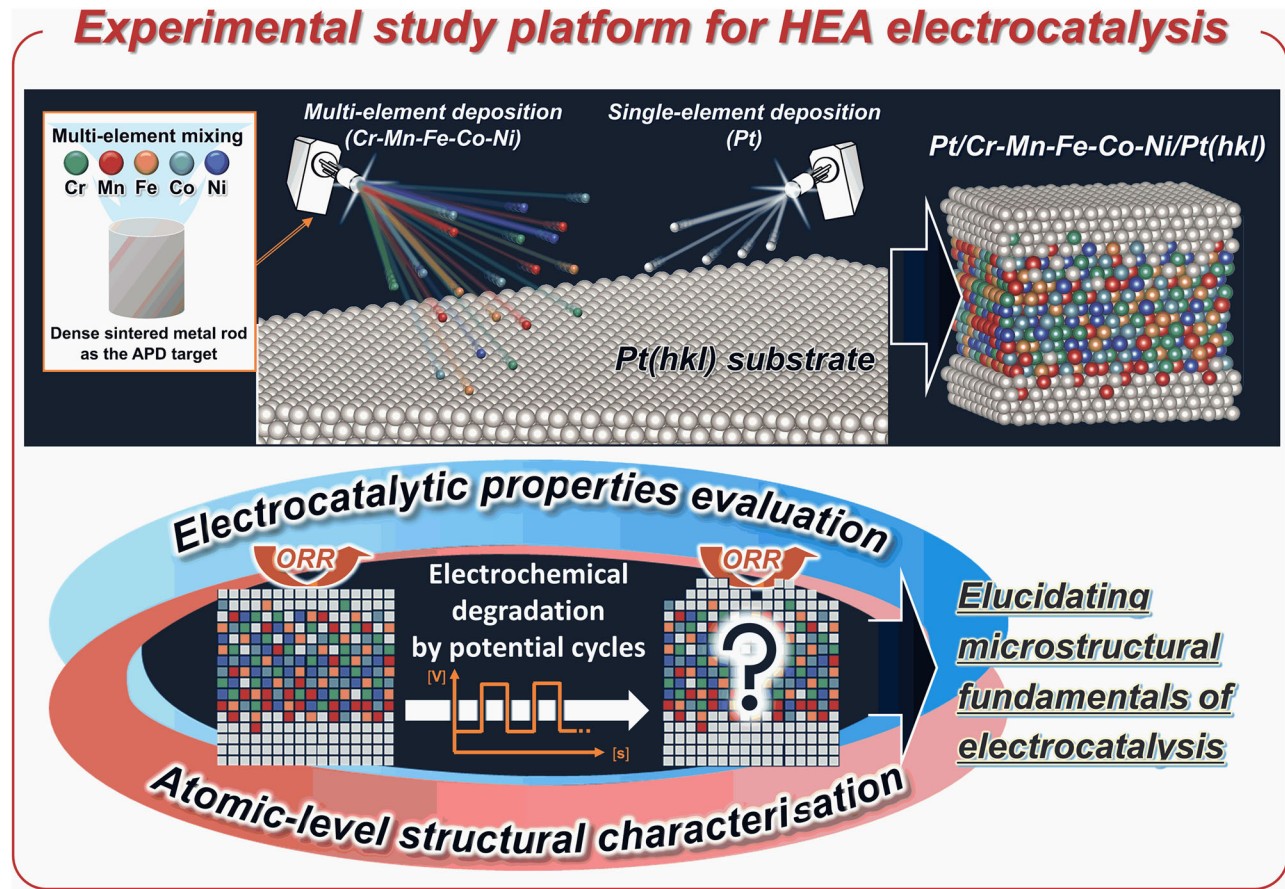

**Fig. 1 | Schematic illustration of the experimental study platform of the HEA electrocatalysis model surface.** Tuning of the elements, mixing ratios of the corresponding APD targets, and alternative and/or simultaneous depositions from multiple APD targets or a combination of the two, enabling complete mixing of HEA deposition with the designed composition. X-ray photoelectron spectroscopy (XPS) was used to confirm that the composition ratio of the resulting HEA layer was highly controlled (Table S1). Characterisation of the atomic-level structure and evaluation of the essential electrocatalytic properties of HEAs by preventing surface oxidation and/or contamination using the sample transfer process are an additional feature of this platform.

crystal surfaces are superior to those of the corresponding Pt-Co binary surfaces (Pt/Co/Pt(hkl)). Furthermore, the properties were determined using the topmost surface atomic arrangements of Pt and vertical elemental distributions near the surface in the depth direction, i.e., the "pseudo-core-shell-like structure" of the underlaid HEA and surface Pt layers, which is attributed to the extraordinary thermodynamic and electrochemical stabilities of HEAs. This work demonstrates that the developed electrocatalysis study platform enables a comprehensive investigation of the typical electrocatalytic (ORR) properties of HEA low-index single-crystal surfaces at the atomic level.

## Results and discussion

### Microstructural observations of the Pt/Cr-Mn-Fe-Co-Ni single-crystal surface vicinities

Because practical ORR alloy catalyst surfaces mainly consist of low-index single-crystal domains of fcc Pt, e.g. (111), (110) and (100), Pt/Cr-Mn-Fe-Co-Ni were vacuum-synthesised (constituent element deposition and thermal annealing) on Pt(hkl) (hkl = 111, 110, 100) substrates and they were labelled as Pt/Cr-Mn-Fe-Co-Ni/Pt(hkl). The atomically-resolved cross-sectional microstructures of the as-synthesised Pt/Cr-Mn-Fe-Co-Ni/Pt(hkl) are shown as corresponding high-angle annular dark field (HAADF) scanning transmission electron microscope (STEM) images in Fig. 2a, c, e. All samples kept lattice-stacking sequences of the Pt(hkl) substrates from the substrates to the topmost surfaces, i.e., the epitaxial growth of both Cr-Mn-Fe-Co-Ni rich lattices and the subsequent surface Pt lattices between the corresponding white dashed lines in each image. One might notice that the Z-contrasts of the STEM images that resulted from the difference in Pt and the Cantor alloy constituent elements depending on the low-index planes of the substrate. Actually, for Pt/Mn-Fe-Co-Ni/Pt(111), the (111) stacking of the Cr-Mn-Fe-Co-Ni-rich lattices (relatively dark area in Fig. 2a) was located between the surface Pt(111) layer and Pt(111) substrate were observed (underlaid bright region in Fig. 2a). Meanwhile, the Cantor alloy constituent elements were widely distributed in the deposition layers of the (110) and (100) samples. Each elemental distribution of the constituent elements in the depth directions shown in Fig. 2b, d, f, which were resolved using energy-dispersive spectroscopy (EDS), validated the difference in the substrate surface lattice-dependent Z-contrasts. Focusing on the deposited layer region (between the dashed lines indicating the topmost surfaces of the samples and dotted lines corresponding to the interfaces between the Pt substrates and the deposited alloy layers), the differences in the resulting elemental distributions were likely due to the thermal-diffusion modes of the constituent elements, which depended on the stacking coordination of the single-crystal structures. In the STEM image of Pt/Cr-Mn-Fe-Co-Ni/Pt(111) (Fig. 2a), a stacking fault is introduced parallel to the stacking plane, which is absent in the images of the (110) and (100) samples (Fig. 2c, e). Otto et al. demonstrated that dislocations introduced into the Cantor alloy tend to be localised in the (111) plane[27,28]. The results suggested that dislocations could be easily introduced out of the parallel to the (110) and (100) stacking planes of the Cr-Mn-Fe-Co-Ni rich lattices. Consequently, faster diffusion of the constituent elements via dislocations occurred in Pt/Cr-Mn-Fe-Co-Ni/Pt(110) and (100) than in Pt/Cr-Mn-Fe-Co-Ni/Pt(111), where body diffusion was the main mode. Notably, the specific vertical distribution of the Cantor alloy constituent elements of Pt/Cr-Mn-Fe-Co-Ni/Pt(111) (Fig. 2a, b) was highlighted by comparing the cross-sectional STEM images and EDS mapping of Pt/Mn-Fe-Co-Ni/Pt(111) (Fig. S4a, b) and Pt/Co/Pt(111) (Fig. S5a, b), which were synthesised under the same conditions except for the APD target materials (an equal composition ratio of Mn, Fe, Co and Ni quaternary alloy, and Co). By comparing the Z-contrasts of the STEM images of the three samples, the thermal diffusions of the Cantor alloy constituent elements into both the surface and substrate Pt(111) stacking lattices were suppressed in Pt/Cr-Mn-Fe-Co-Ni/Pt(111) (Fig. 2b), which implies thermodynamical stability of the Cr-Mn-Fe-Co-Ni (111) stacking layers compared with those of Mn-Fe-Co-Ni or a single element of Co. In

addition, we must point out that by comparing each element, Mn tended to concentrate at the interface regions of the surface and substrate Pt in the Pt/Cr-Mn-Fe-Co-Ni/Pt(111) and Pt/Mn-Fe-Co-Ni/Pt(111) samples. Mn has been reported to have a relatively higher surface energy, higher vapour pressure under vacuum[29], larger body-diffusion kinetics in the Cantor alloy bulk[30] and lower mixing enthalpy with Pt[31] than the other four constituent elements. Therefore, these thermodynamical properties of the constituent elements of the investigated alloys could be closely correlated with the remarkable concentrations of Mn in the interface regions of the Pt lattices.

### Catalytic properties of the Pt/Cr-Mn-Fe-Co-Ni single-crystal surfaces

The ORR properties of Pt/Cr-Mn-Fe-Co-Ni/Pt(hkl) surfaces were evaluated using the electrochemical surface state during the potential cycles (PCs) at 0.6–1.0 V vs. reversible hydrogen electrode (RHE) in an $O_2$ saturated 0.1 M $HClO_4$ solution at 25 °C. The cyclic voltammetry (CV) curves measured in the $N_2$ purged solution at 25 °C are summarised for Pt/Cr-Mn-Fe-Co-Ni/Pt(hkl) before (as-synthesised; blue) and after the 5,000 PC-loading (degraded; grey), as shown in Fig. 3a, c, e. The ORR activity trends evaluated during the 5000 PC-loading are shown in Fig. 3b, d, f. The current density was calculated by dividing the measured current by the geometric surface area defined by the O-ring on the samples. (See details in the Supplementary Information) The CV curves demonstrated typical redox charge features of the Pt-based alloy low-index single-crystal surfaces. For example, the Pt/Cr-Mn-Fe-Co-Ni/Pt(111) surface exhibited a decrease in the adsorption and desorption charges of the H-related species (<0.4 V for Pt(111), 0.13 V for Pt(110) and 0.29 V for Pt(100)). It also simultaneously shifted at the onset potential of O/OH-related species adsorption (>0.6 V) to a higher potential than that of clean Pt(111) (dashed line). The Pt 4f band energy of Pt/Cr-Mn-Fe-Co-Ni/Pt(hkl) evaluated using X-ray photoelectron spectroscopy (XPS) chemically shifted to the high-binding-energy side (Fig. S1g and Table S2), which indicated the so-called ligand effect[32,33], through electron charge transfer from the Cantor alloy constituent elements and/or the strain effect[34] induced by compressive strain of the surface Pt lattice due to the size difference between the surface Pt and the underlaid Cr-Mn-Fe-Co-Ni-rich lattices. Therefore, changes in the adsorption/desorption energy of the H-related and/or O/OH-related species occurred at the topmost surface of the Pt atoms[7,25,35–37].

Figure 3b, d, f show that the ORR activity trends against the PC-loading were significantly different in Pt/Cr-Mn-Fe-Co-Ni/Pt(hkl). The ORR activities of Pt-Co(hkl) and/or Pt-Ni(hkl) single crystal Pt-based model catalyst systems[7,8,38] are typical benchmarks for Pt-based alloy model catalyst studies. Therefore, in this study, evaluated ORR properties for Pt/Cr-Mn-Fe-Co-Ni/Pt(hkl) are discussed comparing with those for Pt/Co/Pt(hkl). The activity of the as-synthesised Pt/Cr-Mn-Fe-Co-Ni/Pt(111) was approximately 12 times higher than that of clean Pt(111), which was probably due to the influence of the ligand effect[32,33] and/or strain effect[34]. The activity monotonically decreased during the PC-loading. The estimated ORR activity loss of Pt/Cr-Mn-Fe-Co-Ni/Pt(111) after 5000 PC-loadings was ~46%, as compared to that of the as-synthesised sample. On the other hand, although the initial ORR activity in Pt/Co/Pt(111) was almost the same (approximately 12 times enhancement), a 47% deactivation was estimated even after 2000 PC-loadings and 5000 PC-loadings resulted in 63% deactivation (Fig. S6b). The ORR activity trends for Pt/Cr-Mn-Fe-Co-Ni/Pt(111) and Pt/Co/Pt(111) were independent of the iR-corrections of the corresponding linear sweep voltammetry (LSV) curves (Fig. S7a), where the corrections were conducted using electrochemical impedance spectroscopy. As shown in the cross-sectional HAADF-STEM images of the degraded Pt/Cr-Mn-Fe-Co-Ni/Pt(111) surface structures (Fig. 4a, b), the surface was composed of Pt(111) epitaxial layers with an average thickness of ~1 nm, and 1–2-nm-high tabletop islands remained on the Cr-Mn-Fe-Co-Ni and/or Pt-Cr-Mn-Fe-Co-Ni layers (relatively dark region). In addition,

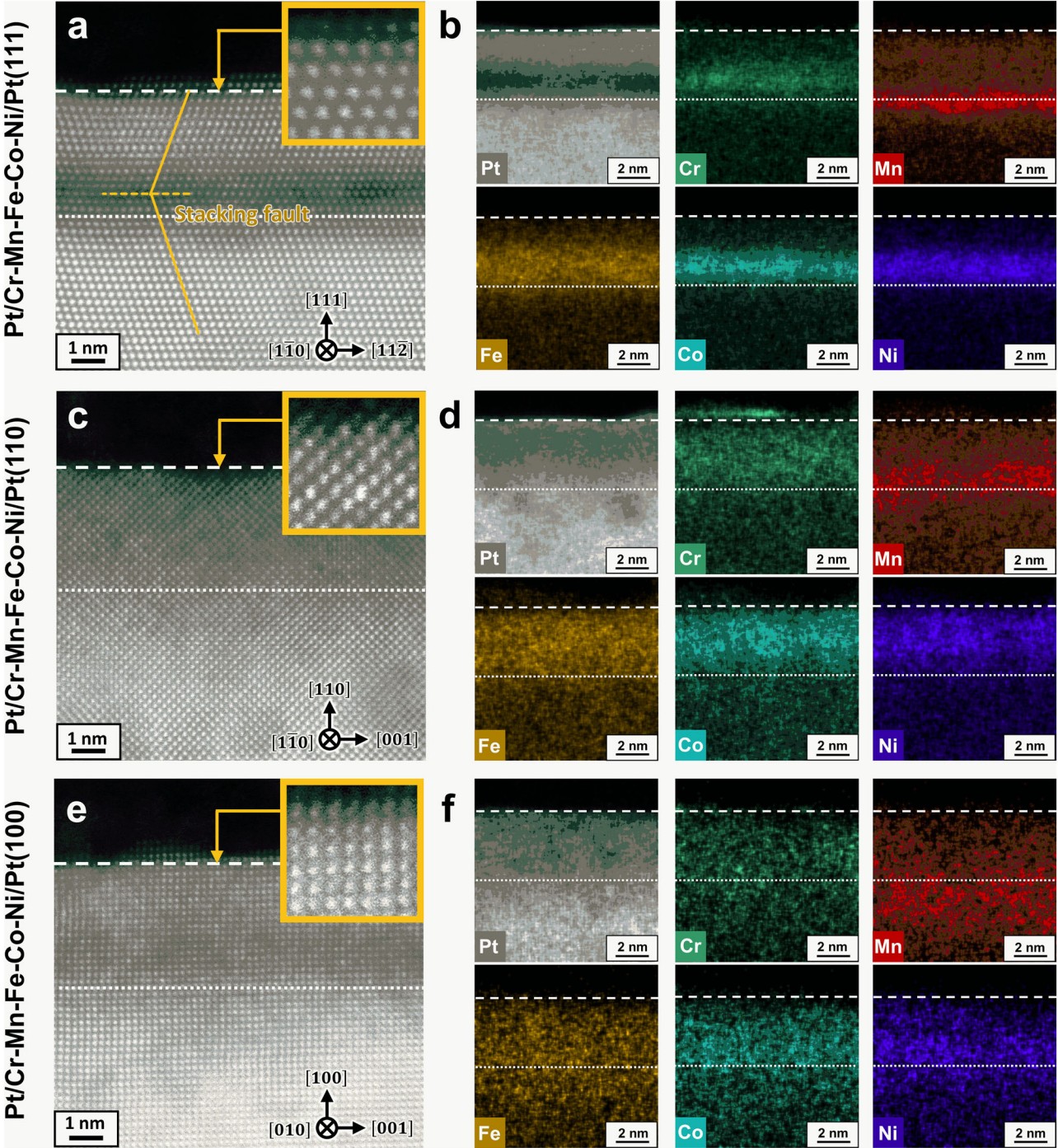

**Fig. 2 | As-synthesised structure characterisations of Pt/Cr-Mn-Fe-Co-Ni/Pt(hkl). a–f** Cross-sectional high-angle annular dark field (HAADF) scanning transmission electron microscope (STEM) images (**a**, **c**, **e**) and colour-coded energy-dispersive spectroscopy (EDS)–2D mapping with a 2-nm-scalebar (**b**, **d**, **f**) of the as-synthesised Pt/Cr-Mn-Fe-Co-Ni/Pt(111), (110) and (100), respectively. Insets in a, c and e show enlarged HAADF-STEM images near each topmost surface. Source data provided for all figures.

the (111) domain size of the topmost surface was within ~1–3 nm although the domain sizes and average surface morphologies in the two observed regions differed (fields of view (FOVs) 1 and 2 in Fig. 4b). Irrespective of the size and orientation changes in the topmost surface Pt domains, the Cr-Mn-Fe-Co-Ni rich interlayer thickness remained nearly unchanged at 5,000 PC-loadings, as determined from the EDS line profiles of the Pt distribution in the depth direction for the as-synthesised (Fig. 4c-1) and 5000 PC-loaded (Fig. 4c-2) samples. This result implies that Pt−Cr-Mn-Fe-Co-Ni was stable against PC-loading. In contrast, the 5,000 PC-loaded and atomically-resolved cross-sectional

STEM images of Pt/Co/Pt(111) (Fig. S5c–e) were considerably different from those of Pt/Cr-Mn-Fe-Co-Ni/Pt(111), in which a relatively rough surface with 3–4 nm-high rounded islands appeared (Fig. S5c, d), which indicates that the surface morphology of Pt/Co/Pt(111) was severely degraded through the underlaid Co dissolution by the PC-loading.

Generally, ORR deactivation of Pt-based alloys proceeds through surface morphological degradations, such as changes in the atomic arrangements and Pt-enriched layer thickness, accompanied by leaching of the constituent less-noble alloying elements located in the surface vicinity[1]. For example, the changes in the atomic location of

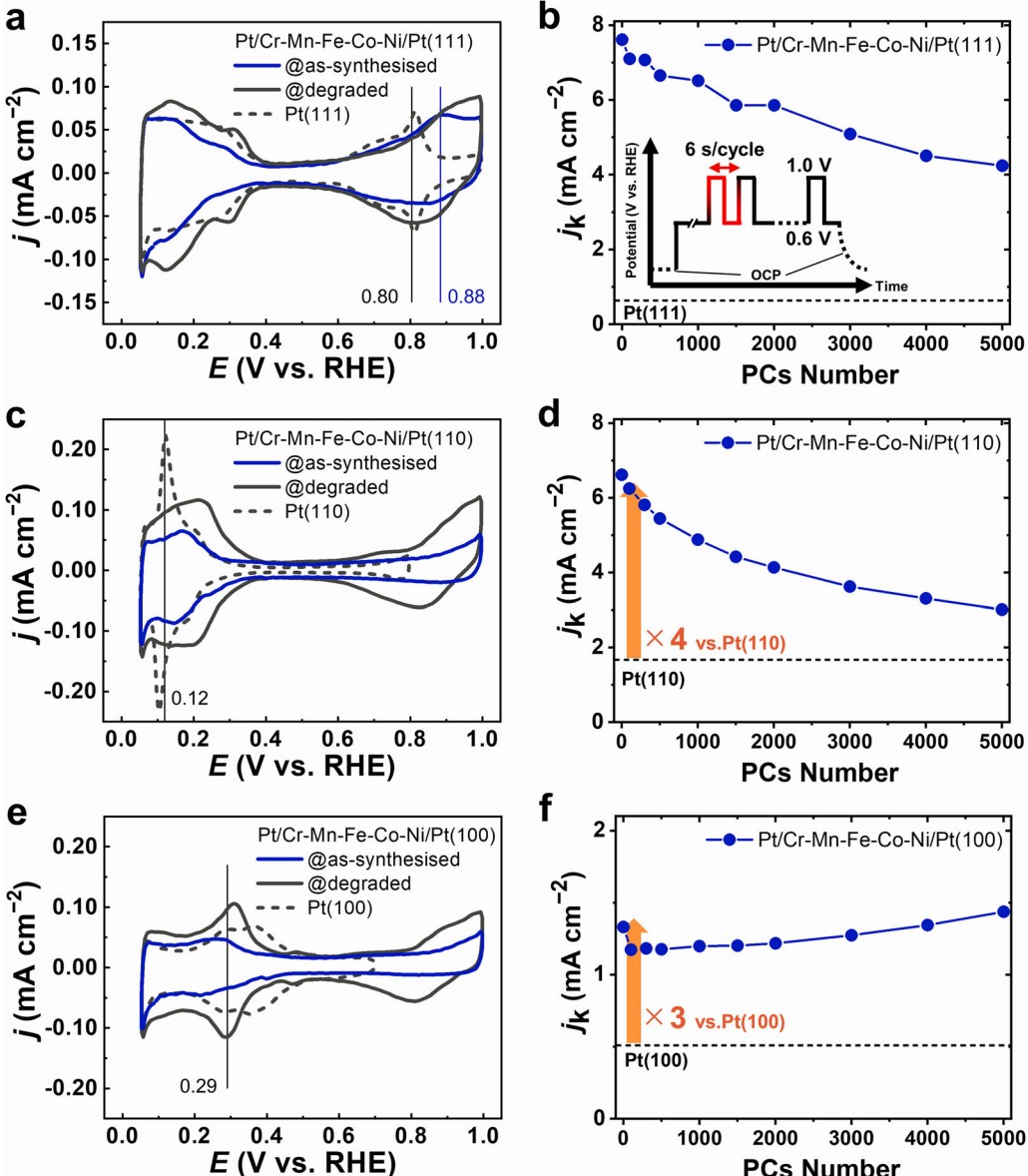

**Fig. 3 | Cyclic voltammetry (CV) curves and ORR activity trends under potential cycles (PC) loading of the vacuum-synthesised Pt/Cr-Mn-Fe-Co-Ni/Pt(hkl).**
**a**–**f** CV curves collected at 0.05–1.0 V vs. reversible hydrogen electrode (RHE) potential range of the as-synthesised (blue) and degraded (grey) states of Pt/Cr-Mn-Fe-Co-Ni/Pt(111), (110) and (100), respectively. Corresponding CVs of the Pt(hkl) substrates are indicated by the dashed lines. **a**, **c**, **e** ORR activity trends recorded under PC-loading (up to 5000 cycles at 0.6–1.0 V) of Pt/Cr-Mn-Fe-Co-Ni/Pt(111), (110) and (100), respectively. **b**, **d**, **f** Insets in **b**: Illustration of a potential cycle protocol for the accelerated degradation test used in this study. (https://www.nedo.go.jp/library/PEFC_CELL_Protocol.html, R−2 III−2−2) Source data provided for all panels.

surface Pt induced by Pt redox (so-called place-exchange) trigger ORR deactivation through atomic-level changes of the topmost Pt surface and alloy compositions in the surface vicinity[39,40]. Therefore, the introduction of additional alloying elements[41,42] and crystallinity tuning[6], both of which optimise the electronic state of the surface elements, have been reported to stabilise the vicinity of Pt-based alloy surfaces. The ORR activity of the densest plane of the fcc single crystal in Pt(111) is known to be enhanced by the ligand and/or strain effects[32–37], which is induced by the underlaid alloying elements of the surface Pt-enriched layer and, thus, such effects rapidly decrease upon applying PC-loading. That is, a comparison between the Pt/Cr-Mn-Fe-Co-Ni/Pt(111) and Pt/Co/Pt(111) single-crystal model catalyst systems is suitable for the discussion of ORR durability depending on the surface structural stability. Also, such the surface structural degradation should be accompanied by an increase in the surface area through atomic-level roughening of the surface Pt layers under the PC-loading condition. According to Fig. S8 in the Supplementary Information, the estimated effective electrochemical surface area (ECSA) for the degraded (after 5000 PCs) Pt/Co/Pt(111), as determined by CO-stripping voltammetry, was obviously larger than that for Pt/Cr-Mn-Fe-Co-Ni/Pt(111) (Fig. S8b), in accordance with the HAADF-STEM findings (Fig. 4 in the main manuscript). The results clearly confirm that the suppression of the surface morphological changes of Pt/Cr-Mn-Fe-Co-Ni/Pt(111) can be linked to limited ORR deactivation (46% after 5000 PC-loadings, estimated based on the geometrical surface area) as compared to Pt/Co/Pt(111) (63%). Moreover, the PC-loading-induced leaching behaviours of the constituent less-noble alloying elements should be responsible for the increase in the electrochemical redox charges of the corresponding CV curves (Figs. 3 and S6), and the effective ECSA estimated by CO-stripping voltammetry for the degraded (5,000 PC-loaded) surfaces (Fig. S8). In fact, the Pt 4$f$ bands shifted to lower binding energies by the PC-loading (Fig. S3).

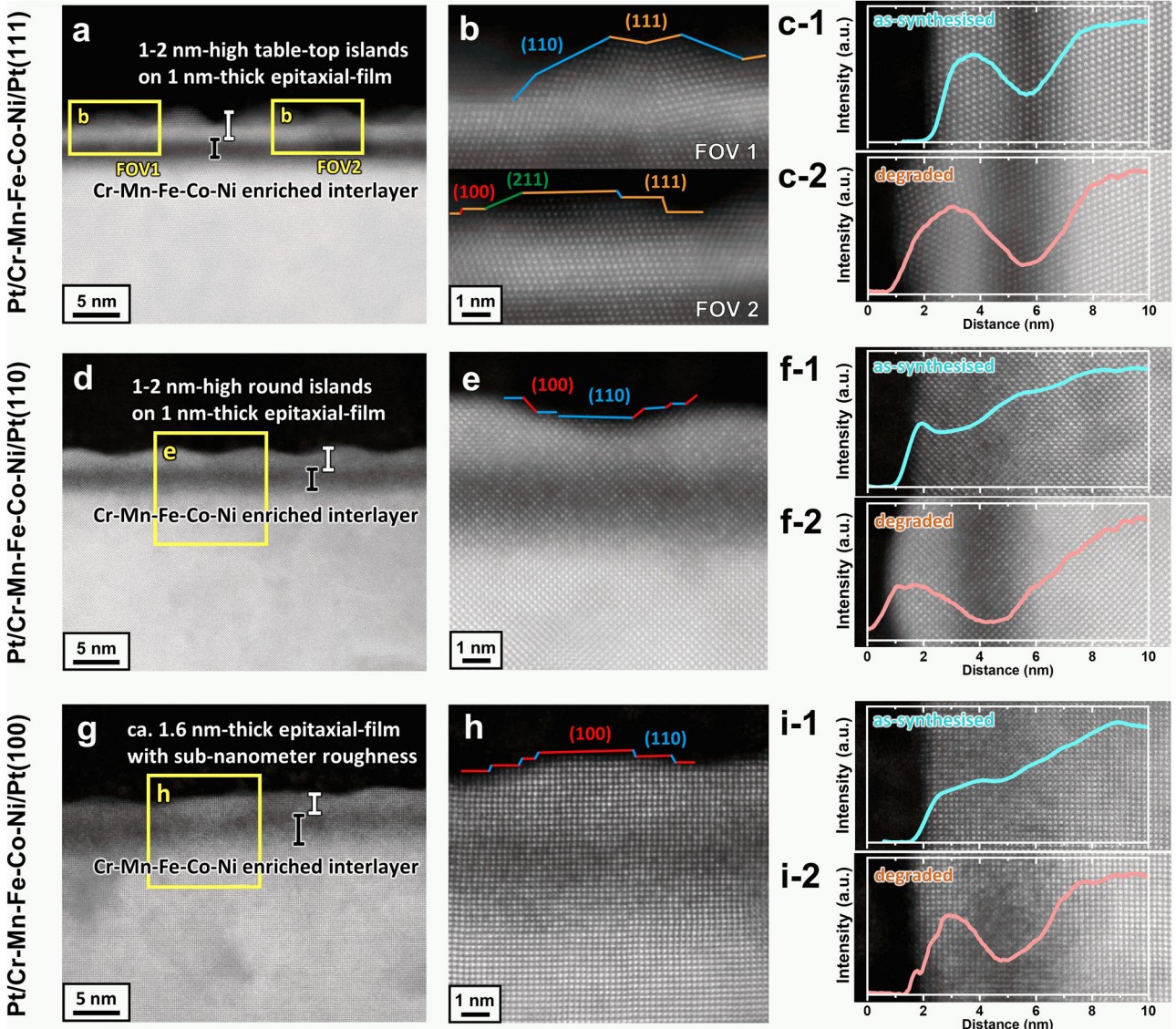

**Fig. 4 | PC-loaded surface degradations of Pt/Cr-Mn-Fe-Co-Ni/Pt(hkl). a–i** Cross-sectional HAADF-STEM images at relatively low magnification [left panels (**a**, **d**, **g**)] and at atomically resolved high magnification [middle (**b**, **e**, **h**)] of the regions marked by the yellow square in **a**, **d** and **g** where the surface domains of (111), (110), (100) and (211) are highlighted in orange, blue, red and green, respectively. EDS intensity depth profiles of Pt in the surface-normal direction of the as-synthesised (1, pale blue) and 5000 PC-loaded (2, pale red) samples of Pt/Cr-Mn-Fe-Co-Ni/ Pt(111), (110) and (100), respectively. [Right panels (**c**, **f**, **i**)] **b** Pt/Cr-Mn-Fe-Co-Ni/Pt (111) shows two field of views (FOVs) to demonstrate the features of the PC-loaded surface. Source data provided for all HAADF-STEM figures and XY-data of the EDS-line profiles.

Pt/Cr-Mn-Fe-Co-Ni/Pt(110) exhibited 4 times higher initial ORR activity compared with vacuum-cleaned pure Pt(110) and remained 2 times enhanced activity after 5,000 PC-loading. (Fig. 3d) As for Pt/Cr-Mn-Fe-Co-Ni/Pt(100), 3 times higher ORR activity than pure Pt(100) were maintained through the PC-loading. (Fig. 3f) The iR-corrected LSVs (Fig. S7b, c) show the similar trends, revealing the out-performed properties of the Pt-HEA system. The ORR activities for the (100) and (110) surface orientations of Pt binary alloy single crystal surfaces have been reported thus far have only undergone slight enhancements against those of pure Pt[7,8,38]. Furthermore, the-oretical calculations revealed that the ligand or strain effects induced by alloying with Co and/or Ni may have a smaller effect on the activity enhancement in the (100) surface orientation than in the (111) surface orientation[43,44]. Actually, Pt/Co/Pt(110) and (100) model catalysts fabricated on this experimental platform showed little or no ORR activity enhancement (Fig. S6d, f). Therefore, the Pt/Cr-Mn-Fe-Co-Ni/ Pt(110) and Pt/Cr-Mn-Fe-Co-Ni/Pt(100) HEA systems surely

correspond to extraordinarily excellent performance in Pt alloy systems. Indeed, the STEM images of the Pt/Cr-Mn-Fe-Co-Ni samples after 5000 PC-loadings exhibited 1–2-nm-high rounded islands with an ~1-nm-thick epitaxial lattice stacking in Pt/Cr-Mn-Fe-Co-Ni/Pt(110) (Fig. 4d, e) and a sub-nanometre order rough surface in Pt/Cr-Mn-Fe-Co-Ni/Pt(100) (Fig. 4g, h). The degraded surface morphologies induced by 5000 PC-loadings corresponded well with the activity trends shown in Fig. 3d, f. Although the effective ECSA for the (110)- and (100)-oriented surfaces increased similarly to the (111)-oriented ones by the PC-loadings, the specific ORR activities that estimated based on the effective ECSA for Pt/Cr-Mn-Fe-Co-Ni/Pt(hkl) were superior to those of Pt/Co/Pt(hkl), in line with the aforementioned discussion. (Fig. S8b, c) Therefore, the outstanding ORR structural stability of the Pt/Cr-Mn-Fe-Co-Ni/Pt(hkl) systems can be explained by the electrochemical stability of the underlaid HEA alloy layers under the oxygen potential of the enriched Pt(hkl) surfaces, as compared to those of Pt/Co/Pt(hkl) binary alloy systems[45–47].

**Table 1 | Values of φ and consisting parameters of the equi-atomic Cr-Mn-Fe-Co-Ni (Cantor alloy) and Pt-Cr-Mn-Fe-Co-Ni alloy**

| HEA (equi-atomic ratio) | $S_c$ (J mol⁻¹ K⁻¹) | $\Delta H_{mix}$ (J mol⁻¹) | $T_m$ (K) | $S_E$ (J mol⁻¹ K⁻¹) | φ |
|---|---|---|---|---|---|
| Cr-Mn-Fe-Co-Ni | 13.4 | −4160 | 1791 | −0.047 | 28.04 |
| Pt-Cr-Mn-Fe-Co-Ni | 14.9 | −11444 | 1833 | −0.091 | 11.49 |

Each value was calculated based on previous studies[48,49]. The $\Delta H_{mix}$ and atomic radius values used in the calculations refer to previous studies[53,54].

Cross-sectional STEM images (Fig. 4a, d, g) of the surface vicinities of Pt/Cr-Mn-Fe-Co-Ni/Pt(hkl) systems show that the Z-contrast induced by the surface Pt-enriched and underlaid Cr-Mn-Fe-Co-Ni rich layers became clearer after the PC-loading, which accompanied the EDS-intensity gaps of Pt (Fig. 4c, f, i). Such PC-loaded structural changes on the surface Pt and underlaid HEA layers were mutually related to the CV redox responses (Fig. 3c, e), where the alloying of Pt with Cantor alloy constituent elements (Fig. 3a, c, e) and Co (Fig. S6a, c, e) exhibit significantly different redoxes due to the H-related (approximately <0.35 V vs. RHE) and O/OH-related (>0.6 V vs. RHE) adsorption and desorption.

Ye et al.[48,49]. proposed parameter φ in Eq. (1) to design HEA.

$$\phi = \left( Sc - \frac{|\Delta H_{mix}|}{T_m} \right) / |S_E| \tag{1}$$

*Sc* denotes the configurational entropy of mixing in an ideal solution. $\Delta H_{mix}$ represents the sum of the mixing enthalpy of the binary liquid between two elements at an equi-atomic composition. $T_m$ is the HEA melting point, which consists of the average values of the melting points of the individual constituent elements and alloy compositions. $S_E$ represents the excessive entropy of mixing, which is a function of the atomic composition, atomic size and overall packing density. For example, the crystal phases of HEA can be predicted using the value of φ. HEAs with φ that is <3.5 exhibit an amorphous structure, that of more than 20 creates a single-phase solid solution and that between 3.5 and 20 corresponds to a multiphase alloy composed of solid solutions, intermetallic compounds, or both. The values of φ in Cr-Mn-Fe-Co-Ni (Cantor alloy) and in the equi-atomic ratio Pt-Cr-Mn-Fe-Co-Ni are 28 and -11.5, respectively, as listed in Table 1. The estimated φ value of Pt–Cr-Mn-Fe-Co-Ni indicated that the crystal phase of the six-element alloy is not a single-phase solid solution but a multiphase alloy. The difference in the crystal phases could be a 'driving force' for the PC-loaded phase separation behaviour. A significant change in φ could result in the 'pseudo-core-shell like' structure that consists of surface Pt-rich and underlaid Cr-Mn-Fe-Co-Ni-rich layers and, thereby, the corresponding phase separation behaviour of the six-element single-crystal alloy surfaces of Pt-Cr-Mn-Fe-Co-Ni synthesised in this study. Although, at present, no definitive explanation can be made for the phase separation behaviour of the surface Pt and underlaid Cr-Mn-Fe-Co-Ni lattices, the PC-loading-induced phase separations accompanied by surface morphological changes not only of (111) but also of (100) and (110) surfaces should determine the outperformed ORR properties of Pt-HEA systems through the formations of the "pseudo-core-shell-like structure" induced by the PC-loading. Such unique phase separation behaviour of the Pt/Cr-Mn-Fe-Co-Ni HEA system under electrochemical PC-loading demonstrates the material potential of Pt-Cr-Mn-Fe-Co-Ni hex-ternary HEA as a next-generation PEMFC cathode catalyst.

We developed an experimental electrocatalysis study platform that enables the synthesis of well-defined single-crystal HEA model catalyst surfaces as well as typical electrocatalysis (ORR) of the resulting Pt/Cr-Mn-Fe-Co-Ni HEA system was investigated through atomic-level evaluations of the surfaces. Consequently, well-controlled single-crystal Pt/Cr-Mn-Fe-Co-Ni surfaces were successfully vacuum-synthesised. A detailed microstructural analysis of the surfaces before and after the electrochemical evaluations demonstrated that the atomic-level microstructural factors that contributed to the ORR properties could be traced. In particular, the elemental distributions of the Cr-Mn-Fe-Co-Ni HEA layers determined the ORR properties, particularly the electrochemical structural stability of the Pt-enriched surface layers. Furthermore, it was found that the PC-loading of Pt/Cr-Mn-Fe-Co-Ni/Pt(hkl) hex-ternary alloy surfaces generated the "pseudo-core-shell-like structure" comprised of the Pt-enriched surface and the underlaid Cr-Mn-Fe-Co-Ni rich layers. This study validated the experimental study platform for clarifying the precise correlations between the atomic-level surface microstructure and electrocatalytic properties of HEAs at any constituent elements and ratios. The use of this platform can bridge the gaps between the scientific computational predictions provided by MI and practical catalytic properties of nanosized HEA catalysts, through providing a reliable dataset: this powerful experimental platform is applicable not only to electrocatalysis but also in various fields of HEA functional nanomaterials.

## Methods

### UHV chamber for the synthesis of atomical structure-controlled HEA surfaces

Sample fabrication was performed in a UHV chamber equipped with the following instruments: a manipulation stage with pyrolytic graphite heaters, an ion source gun (PSP; ISIS3000) for surface cleaning (Ar⁺ sputtering), multiple APD sources (ADVANCED RIKO; APS-1) and a quartz-crystal microbalance (Sigma Instruments: SQM-160) to control the deposition rate. High-purity metal or alloy rods were used as the APD source targets. The Cr-Mn-Fe-Co-Ni layer in this study was synthesised from a custom-made target (>99.9% overall purity; atomic ratio of Cr-Mn-Fe-Co-Ni = 1.0:1.0:1.0:1.0:1.0) (Toshima Manufacturing Co., Ltd., Material System Division), which was sintered by mixing Cr, Mn, Fe, Co and Ni in equal composition ratios. Pure Pt (>99.95% purity; The Nilaco Corporation) APD targets were commercially available. Equi-atomic quaternary Mn-Fe-Co-Ni alloy (> 99.9% overall purity; atomic ratio of Mn:Fe:Co:Ni = 1.0:1.0:1.0:1.0; Toshima Manufacturing Co., Ltd., Material System Division) and pure Co (> 99.95% purity; The Nilaco Corporation) APD targets were used as reference samples for synthesis using the same procedure as the Cr-Mn-Fe-Co-Ni target.

### Vacuum synthesis procedure

The Pt(hkl) (hkl = 111, 110, 100 | MaTeck; φ = 10 mm, t = 1 mm, <0.1° miscut) crystal surfaces were cleaned using repeated Ar⁺ sputtering and annealing at 1273 K under UHV. Pt, Cr-Mn-Fe-Co-Ni, Mn-Fe-Co-Ni and/or Co were arc-plasma-deposited on Pt(hkl) at a bias arc voltage of 70 V and pulse frequency of 2 Hz. The deposition thickness of each element was determined using a quartz-crystal microbalance installed in the UHV chamber. By considering the atomic radii, the estimated mass thickness of 0.3 nm was calibrated to be one-monolayer (ML) thick. Following a 10 ML-thick Cantor alloy constituent was deposited on the cleaned surface of Pt(hkl) at 300 K, the samples were annealed under UHV at 773 K for 30 min. Then, a 4 ML-thick Pt layer was deposited on the pre-deposited layer at 300 K and annealed at 623 K. The synthesised samples were designated as Pt/Cr-Mn-Fe-Co-Ni/Pt(hkl). Pt/Mn-Fe-Co-Ni/Pt(hkl) and Pt/Co/Pt(hkl) were also synthesised for comparison. Clean Pt(hkl), which was also used as a reference in the electrochemical evaluations, was prepared using Ar⁺ sputtering and annealing only.

### Surface structural characterisation

XPS analysis of the as-synthesised surfaces was performed in another UHV chamber using a semi-spherical electron energy analyser (PSP;

RESOLVE 120 MCD5) and a Mg Kα X-ray source (PSP; CTX400). The sample was transferred from the sample-preparation chamber to the XPS chamber without air exposure, using a vacuum-transfer vessel to prevent surface oxidation and contamination. Atomically-resolved surface cross-sectional images and elemental maps of the samples were obtained using a combination of HAADF-STEM and EDS (JEOL; JEM-ARM200F and FEI Company; TITAN$^3$ G2 60−300 installed Hitachi, S-5500, respectively). The samples for STEM-EDS observations were prepared after using C coating and Ga focused ion-beam process on the surface vicinity of the sample.

## Electrochemical (ORR) property evaluation

The electrochemical measurements were performed using a conventional three-electrode electrochemical cell. RHE ($H_2$ gas-flow type, open compartment design[50]) and Pt wire were used as the reference and counter electrodes, respectively. Electrochemical measurements of the catalyst surface models were performed using a potentio-galvanostat (HZ-5000, Hokuto Denko) combined with a rotating-disc electrode system (HR-301, Hokuto Denko). The experimental environment is temperature-controlled at a constant 25 °C by air conditioning. Following the vacuum-transfer using the vacuum-transfer vessel, the geometrical sample surface area was regulated using a Karletz O-ring (Dupont-P4) in a $N_2$-purged glove box to prevent surface oxidation. To maintain the atomic arrangements of the UHV-fabricated sample surfaces, the first CV curves were collected by immersing the sample working electrodes (sample surfaces) in $N_2$-purged 0.1 M $HClO_4$ under holding a constant potential of 0.08 V vs. RHE at 25 °C, i.e., without undergoing open circuit potential (approximately 0.8 V vs. RHE). A scan rate of 0.05 V/s was used to acquire the CV curves in the potential range of 0.05−1.0 V without sample-disc rotation until the CV shape was stabilised (approximately ~10 cycles). Subsequently, LSV was conducted at various disc-rotation rates of 400−2500 rpm at a positive potential-sweep rate of 10 mV/s in the potential range of 0.05−1.05 V for ORR activity evaluations after $O_2$ saturation of the solution. After CV collection, the sample electrodes were removed from the $N_2$-purged solution and the surface was protected in $N_2$-bubbling ultrapure water to avoid possible oxidation of the surface while the solution was saturated with $O_2$. The ORR activities were evaluated using the kinetic current density ($j_k$), which was estimated using the Koutecky-Levich equation[51]. $j_k$ values of 0.9 V vs. RHE were used as a criterion for the ORR activity. iR correction was generally not performed except as specified. The electrode surface stabilities, i.e., evaluations of ORR durability, were investigated by applying square-wave PC-loading between 0.6 and 1.0 V every 3 s in an $O_2$ saturated solution at 25 °C[52], as shown in the inset of Fig. 3b. Carbon monoxide (CO) stripping voltammetry was conducted to estimate the effective ECSAs of the sample surfaces prepared using the same vacuum-synthesis procedure as for Pt/Cr-Mn-Fe-Co-Ni/Pt(hkl) and Pt/Co/Pt(hkl). The voltammetry procedure is as follows: the potential of the working electrode keeps at 0.08 V vs. RHE during an immersion of the vacuum-synthesised sample surface into $N_2$-purged 0.1 M $HClO_4$ solution, CO was then introduced into the solution until the adsorbed CO reached saturation on the sample surfaces, followed by re-purging of the solution by $N_2$ to remove the excess dissolved CO. Then, CV was collected in the range 0.05−1.0 V, started with positive-going sweep.

## Data availability

All data supporting the findings of this study have been included in the main text and Supplementary Information. All additional materials and data are available from the corresponding author upon request. Our study did not use any custom code or mathematical algorithm. XY-data concerning the electrochemical properties (Figs. 3 and S6) and the EDS signal intensity of Pt (Fig. 4), and separated raw images of Figs. 2 and Fig. 4 are provided in this Source data with this paper. Source data are provided with this paper.

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

## Acknowledgements

This research was supported by the New Energy and Industrial Technology Development Organization (NEDO) of Japan, JPNP20003 (W.T.), JSPS KAKENHI, Grant Number JP21H01645 (W.T.), JST SPRING, Grant Number JPMJSP2114 (Y.C.) and Grant-in-Aid for JSPS Fellows, Grant Number JP23KJ0111 (Y.C.).

## Author contributions
Y.C. conceived and coordinated all stages of this research. T.T. and T.E. was involved in investigation, review and editing. Noboru T. conducted STEM-EDS analysis and data processing. T.I. was involved in review and editing. K.H. and Naoto T. were involved in review and editing. T.W. was involved in conceptualisation, funding acquisition, project administration, supervision, review and editing. All authors reviewed the manuscript and approved the final report.

## Competing interests
The authors declare no competing interests.

### Ethical approval
We affirm that we respect the "Ethics & Inclusion statement" of Nature Communications and that we have based our research plans and activities on this statement.
