## [Peer Review File · Nature Communications]

Reviewer comments first round

Reviewer #1 (Remarks to the Author):

Comments:

In this manuscript, the authors fabricated the nanometer-thick epitaxial stacking layers of Pt and Cr-Mn-Fe-Co-Ni on low-index single-crystal Pt substrates. Then, based on the designed platform, the relationship between the multicomponent alloy surface microstructures and their catalytic properties was studied through the combinations of experimental studies and materials informatics. As illustrated in the submitted manuscript, the atomic-level microstructural factors could optimize single-crystal Pt/Cr-Mn-Fe-Co-Ni surfaces to further tune its electronic and overall catalytic properties. Besides, the elemental distributions of the Cr-Mn-Fe-Co-Ni HEA layers could influence the electrochemical structural stability of the Pt-enriched surface layers. However, the relationship between the surface microstructures and their catalytic properties were not systematically studied, and the materials informatics did not provide convinced interpretation for the practical catalytic properties of nanosized HEA catalysts. According to the above reasons, this manuscript is not recommended for publication in Nature Communications.

Reviewer #2 (Remarks to the Author):

I have read the manuscript "Experimental study platform for electrocatalysis of 3 atomic-level controlled high-entropy alloy surfaces". The manuscript describes the synthesis, characterization and electrochemical testing of a stacked Pt/CrMnFeCoNi HEA/Pt. I find the material and method interesting, however, there is a number of questions and issues I would like to raise:

1. Page 4: The authors claim they conduct the HEA synthesis under UHV $<10^{-7}$ Pa and transfer it directly to an electrochemical cell, partially, to avoid exposure to air. However, they test the catalyst for oxygen reduction, where they need to saturate the electrolyte with oxygen. Do they transfer the electrode under potential control? If not, what is the point of avoiding exposure to air if the electrode is later exposed to oxygen anyway?
2. Page 5 The authors compare the catalyst activity to the activity of a stacked Pt/Co/Pt catalyst. What is the meaning of this comparison? PtCo alloys are indeed some of the best ORR catalysts, but in a nanoparticulate Pt-Co core-shell structure. the Pt/Co/Pt catalyst mentioned here is a completely different material.
3. Figure 2: in the 2D EDS images all elements seemed to have diffused to all areas of the catalyst layers. How were these cross-sectional images obtained practically? Was the sample cut or polished or in any other way modified? What are the chances of cross-contamination between layers due to this?
4. Figure 3: All synthesized catalysts have an increased Hads/des region after cycling indicating significant changes in the alloy surface. How did the authors normalize the currents? how did they determine the electrochemical surface area? Additionally, the authors must show actual ORR polarization curves, it is quite unusual not to display them prominently in an electrocatalysis manuscript. Have the potentials been ORR corrected? How was the uncompensated resistance determined? Please show the EIS spectra. All these factors are crucial for understanding HEA activity, as it has been outlined (e.g. ChemEngineering 2022, 6, 19; ACS Catal. 2021, 11, 1014–1023; Appl. Surf. Sci. 2020, 533, 147471).
5. The authors repeatedly use the term "PC-loading" and "potential cycles loading". Is this the same as regular cyclic voltammetry? As far as I understand yes - then please use standard terminology. If not, please elaborate the difference.
6. Page 16: The authors claim that their system is favorable for practical applications compared to

the PtCo system. Considering the lack of stability of the presented catalysts I consider this claim needs to be more substantially justified, or removed.

7. Page 18: the authors claim that the catalysts display a core-shell structure, however the structure they observe is not what is classically called a core-shell structure (Nat. Chem. 2010, 2, 454–460; Nat. Mater. 2007, 6, 241–247; Rev. Chem. Eng. 2009, 25, ; J. Electroanal. Chem. 2003, 554–555, 191–199; J. Am. Chem. Soc. 2006, 128, 8813–8819)

Reviewer #3 (Remarks to the Author):

In the manuscript the authors report the preparation of atomic-level controlled platinum high-entropy alloy (HEA) surfaces on substrate using arc plasma deposition method and the study of the electrocatalytic properties in oxygen reduction reaction (ORR) to demonstrate this APD method for preparing experimental study platform of HEAs. Overall I think this is a good quality paper. HEAs have attracted considerable research interest, with many exotic properties reported but the mechanisms not well elucidated. This work provides a method to finely control HEA composition and surface exposure, which can serve as model structure in fundamental studies. I think it can be considered for acceptance after the following questions are addressed.

1) It is very interesting the HEA layer as well as Pt overlayer are epitaxially deposited, i.e., with the surface plane remaining the same as the substrate surface plane. Considering arc plasma ions carries relative high kinetic energies, I would assume the ions would sputter the substrate during deposition process and the deposition process would be dominated more by deposition kinetics rather than thermodynamics, which thus would cause surface roughness. Would the authors elaborate more regarding the epitaxial deposition mechanism?

2) The Pt/HEA layer surface became rough after ORR stability test. The authors attributed this to significant element leaching as well as element segregation in the as-prepared Pt/HEA layer. I was wondering if HEA elements were already in the as-prepared Pt overlayer and there was significant strain in the as-prepared Pt/HEA layer, which could also contribute to the Pt surface roughing during CV cycling in ORR test condition. Would the authors comment on these possibilities?

Replies to Reviewers # NCOMMS-22-38382A

Thank you for your time and effort in reviewing our submitted manuscript. We appreciate the reviewers' constructive suggestions and comments. We have carefully considered the reviewers' comments and revised the main and Supplementary Information manuscripts accordingly. We have also prepared point-by-point responses to each comment below. Some of the wording has been revised in the main and Supplementary Information manuscripts.

Note: The revised parts are highlighted (yellow) in the **marked-up** main and Supplementary Information manuscripts. The highlighted parts in the manuscripts show the corresponding comments from the reviewers. For example, the part corresponding to comment 1 by reviewer 2 is designated as [R.2(1)]. In the following text, the comments from the reviewers are presented in blue. Additionally, we provide the locations to the changes in the manuscripts in the all responses as *P.* and *l.*, for example *P. 1 l. 6-12*, which refers to the page and line numbers.

Reviewer #1 (Remarks to the Author):

Comments: In this manuscript, the authors fabricated the nanometer-thick epitaxial stacking layers of Pt and Cr-Mn-Fe-Co-Ni on low-index single-crystal Pt substrates. Then, based on the designed platform, the relationship between the multicomponent alloy surface microstructures and their catalytic properties was studied through the combinations of experimental studies and materials informatics. As illustrated in the submitted manuscript, the atomic-level microstructural factors could optimize single-crystal Pt/Cr-Mn-Fe-Co-Ni surfaces to further tune its electronic and overall catalytic properties. Besides, the elemental distributions of the Cr-Mn-Fe-Co-Ni HEA layers could

influence the electrochemical structural stability of the Pt-enriched surface layers. However, the relationship between the surface microstructures and their catalytic properties were not systematically studied, and the materials informatics did not provide convinced interpretation for the practical catalytic properties of nanosized HEA catalysts. According to the above reasons, this manuscript is not recommended for publication in Nature Communications.

Reply: First, we aimed to construct a powerful experimental platform, particularly for electrocatalysis studies of multi-component alloy systems, such as HEAs, because the chemical structures of catalyst surfaces are significantly complicated owing to the constituent elements, compositions, and atomic arrangements. As described in the main manuscript, Pt/Cr-Mn-Fe-Co-Ni/Pt(*hkl*) model catalyst surfaces were synthesised under UHV conditions using the arc-plasma deposition method of Cantor alloy and Pt, and atomic-level controlled surface structures were demonstrated using STEM-EDS images. Then, the ORR properties of the model catalyst surfaces were evaluated using electrochemical measurements. The results demonstrated outperformed properties, as compared to the benchmark model catalyst surfaces of Pt/Co/Pt(*hkl*) binary alloy system. This study confirmed the effectiveness of the experimental platform by clarifying the atomic-level surface structures and unique ORR properties.

> The relationship between the surface microstructures and their catalytic properties were not systematically studied.

We would like to emphasise that catalysis studies using single-crystal surfaces as a model catalyst surface is a powerful experimental approach. Such model catalyst surfaces provide information on the fundamental aspects of the correlation between the atomic arrangements of the catalyst surface and the corresponding catalytic activity. Practical

electrocatalysts, such as catalysts for PEMFC, are nanoparticulate-shaped to increase the catalytically-active areas for target reactions such as the ORR. Their catalytic properties are generally determined by various structural factors, such as particle size distribution, shape, dispersion state on the support material and compositional heterogeneity. The single-crystal model catalyst alloy surfaces enable the definition of at least the surface atomic arrangements and elemental distributions near the topmost surface vicinity and, thereby, clarify the potential of HEAs as electrode catalysts.

For example, we found that electrochemical PC-loading structurally enhances the surface-enriched Pt and underlaid Cantor alloy-rich phase separation. We described this as a pseudo-core-shell structure. (Related also to the comment for Reviewer 2 #7) Such a phase separation behaviour is an essential material property of Pt-HEAs and determines the outperformed ORR properties, as compared to the Pt-Co benchmark binary alloy surface system.

Moreover, as shown in Figs. S6d and S6f, although the ORR activity enhancements of Pt/Co(Ni)/Pt(110) and (100) vs. the corresponding pure Pt(110) and (100) single-crystal surfaces are only slight (please also see Y. Yamada *et al.*, *Surf. Sci.*, **607** (2013) 54-60; S. Kobayashi *et al.*, *J. Phys. Chem. C*, **121** (2017) 11234-11240 (Pt-Co) and V. R. Stamenkovic *et al.*, *Science*, **315** (2007) 493-497 (Pt-Ni)), and the theoretical calculation results of Z. Duan and G Wang, *J. Phys. Chem. C*, **117**, 12 (2013) 6284-6292 and K. Li *et al.*, *J. Mater. Chem. A*, **3** (2015) 11444-11452), Pt/Cr-Mn-Fe-Co-Ni/Pt(110) and (100) exhibited enhancement factors of 3–4, demonstrating the unique catalytic properties of Pt-HEA multi-component alloy surfaces, as compared to benchmark Pt-Co(Ni) binary surfaces. The results also demonstrate the outstanding material potential of the Pt-HEA multi-component alloy system and the effectiveness of our experimental platform.

> *The materials informatics did not provide convinced interpretation for the practical catalytic properties of nanosized HEA catalysts.*

Our paper does not contain the Materials Informatics (MI) results, but the detailed relationship between the ORR properties and surface atomic-level microstructures of Pt/Cr-Mn-Fe-Co-Ni/Pt(*hkl*) model catalyst surfaces is discussed using our experimental platform. Considering the versatility of so-called “HEAs” (infinite combinations of constituent elements and compositions), the MI approach may be crucial for improving the efficiency of selecting suitable materials. We believe that our experimental data on multi-component alloys, particularly HEAs, obtained using the model catalysts and the experimental platform are very useful as reliable training datasets for MI. In fact, our research group has started searching for the best HEA system for superior ORR properties by using active learning.

The insights provided in this study using Pt-HEA single crystal surfaces are very valuable for practical novel catalyst development, that is, nanoparticles of Pt-HEA. For example, truncated octahedron (TOC) nanoparticles of face-centred-cubic (fcc) metals or alloys and UHV-synthesised model catalyst surfaces with (111), (100) and (110) orientations correspond to the terrace (111, 100) and edge/corner (110) regions of the TOC nanoparticles. The ORR properties of various shape-controlled nanomaterials (octahedron, nanorods, nanosheets and others) have been investigated, so that the most active (111) surface is widely exposed (B. R. Cuenya, *Thin Solid Films*, **518**, 12 (2010) 3127-3150; Y. Wang *et al.*, *Energy Environ. Sci.*, **11** (2018) 258-275; M. Liu *et al.*, *Adv. Mater.*, **31** (2019) 1802234 1-8; and E. Hornberger *et al.*, *ACS Appl. Energy Mater.*, **4** (2021) 9542-9552). Such the studies have been conducted on the basis of experimental results obtained for the single crystal surfaces.

Although various HEA catalysts certainly show promising catalytic properties (ORR in Alkali medium: T. Loffler *et al.*, *Adv. Energy Mater.*, **8**(34) (2018) 1802269 1-7; EOR: D. Wu *et al.*, *JACS*, **142** (2020) 13833-13838 and CO₂RR: J. K. Pedersen *et al.*, *ACS Catal.*, **10** (2020) 2169-2176), the reported HEA catalyst studies are still in progress and practical catalysts are in a very early stage of development. We are confident that the fundamental findings of the Pt-HEA model catalyst surfaces presented in this manuscript will provide new insights into HEA catalyst applications not only for ORR, but also for various electrochemical reactions, such as OER, CO₂ reduction and NH₃ synthesis.

We have revised the sentence structures and data presented throughout the manuscript by adding a discussion to clarify our views. (P.4 l.6-12; P.5 l.4-6, l.19-21; P.15 l.9-P.16 l.6; P.16 l.7-l.20; P.17 l.2-5; P.18 l.14-20) We hope the revised version will be reviewed again.

Reviewer #2 (Remarks to the Author):

I have read the manuscript "Experimental study platform for electrocatalysis of 3 atomic-level controlled high-entropy alloy surfaces". The manuscript describes the synthesis, characterization and electrochemical testing of a stacked Pt/CrMnFeCoNi HEA/Pt. I find the material and method interesting, however, there is a number of questions and issues I would like to raise:

1. Page 4: The authors claim they conduct the HEA synthesis under UHV ($<10^{-7}$ Pa) and transfer it directly to an electrochemical cell, partially, to avoid exposure to air. However, they test the catalyst for oxygen reduction, where they need to saturate the electrolyte with oxygen. Do they transfer the electrode under potential control? If not, what is the point of avoiding exposure to air if the electrode is later exposed to oxygen anyway?

Reply: Our experimental platform enables a detailed discussion of the electrochemical properties of well-defined (surface atomic arrangements and vertical elemental distributions in the surface vicinity) UHV-synthesised Pt-HEA(*hkl*) surfaces based on cyclic voltammetry (CV) and linear sweep voltammetry (LSV). It is well known that the atomic arrangements of UHV-cleaned single-crystal surfaces are disrupted when they are exposed to air. We evaluated the essential ORR properties of clean, well-defined Pt-HEA(*hkl*). Thus, the CV and the first LSV before PC-loading curves must be acquired for the preserved surface structures by the transport of the as-synthesised sample under UHV to the electrochemical measurement environments.

Specifically, CV should be carefully conducted. In this study, the UHV-synthesised Pt-HEA(*hkl*) samples were transferred from the UHV chamber to an N₂-purged 1 atm glove-box using a sample transfer system (please see for example, T. Wadayama *et al.*, *Electrochem. Commun.* **12**, 8 (2010) 1112-1115). After placing the sample in the glove-box, the surface of the sample substrate (working electrode) was immersed in an N₂-purged 0.1M HClO₄ solution at a constant applied potential of 0.08 V vs. RHE without undergoing open circuit potential (OCP). These procedures were aimed at preventing possible oxidation of the well-defined surfaces. Furthermore, the potential window of CV to limit electrochemical oxidation of the surface Pt-enriched layers was typically 0.05–1.0 V. After the CV measurements, the sample electrodes were removed from the solution and their surfaces were protected by placing them in N₂-purged Milli-Q water. Then, they were re-immersed in an O₂-saturated solution at a constant potential of 0.6 V vs. RHE, and LSV measurements were immediately started at each disk rotation speed.

Descriptions of the sequences for the electrochemical experimental procedures adopted in this study have been added to the Methods section of the revised manuscript. (P.23 l.4-9, l.13-16)

2. Page 5 The authors compare the catalyst activity to the activity of a stacked Pt/Co/Pt catalyst. What is the meaning of this comparison? Pt-Co alloys are indeed some of the best ORR catalysts, but in a nanoparticulate Pt-Co core-shell structure. The Pt/Co/Pt catalyst mentioned here is a completely different material.

Reply: In this study, we used Pt/Co/Pt(*hkl*) samples, which were synthesised using similar experimental procedures (described in P.20 l.17–P.21 l.9 of the Methods section in the

main manuscript), as a benchmark single-crystal surface structure for the ORR study. As pointed out by the reviewer, Pt/Co/Pt(*hkl*) is a stacking structure of surface Pt-enriched and underlaid Co-rich layers on a Pt(*hkl*) substrate, and the structures are different from those of nanosized practical catalysts. However, such stacking layers can be considered as one of the catalyst surface models for Pt-Co core-shell type structures, where the Pt-Co alloy core is located underneath the Pt-enriched shell of practical catalysts (Fig. S5b in the Supplementary Information).

Many studies have reported to date for the ORR properties of binary alloy model single-crystal catalysts such as Pt-Co and Pt-Ni (please see for example, V. R. Stamenkovic *et al.*, *Science*, **315** (2007) 493-497; Y. Yamada *et al.*, *Surf. Sci.*, **607** (2013) 54-60 and S. Kobayashi *et al.*, *J. Phys. Chem. C*, **121** (2017) 11234-11240; etc.). Therefore, we considered that a comparison of the ORR properties of UHV-synthesised Pt/Cr-Mn-Fe-Co-Ni/Pt(*hkl*) and Pt/Co/Pt(*hkl*) benchmark surfaces are the best way to highlight the superior ORR properties of Pt/Cr-Mn-Fe-Co-Ni/Pt(*hkl*) and to emphasise the material potential of Pt-Cr-Mn-Fe-Co-Ni HEA alloys for practical ORR catalysts.

We have added some sentences to the manuscript to address this issue. (P.14 l.2-5; P.16 l.7-20)

3. Figure 2: in the 2D EDS images all elements seemed to have diffused to all areas of the catalyst layers. How were these cross-sectional images obtained practically? Was the sample cut or polished or in any other way modified? What are the chances of cross-contamination between layers due to this?

Reply: Cross-sectional STEM-EDS observations and analyses were conducted on samples prepared using the conventional technique, whereby the sample surface of the UHV-synthesised single-crystal model catalyst was coated with carbon paste and then cut out using a Ga focused ion-beam (FIB).

In the EDS-2D mappings (Figs. 2b, 2d and 2f), the following factors may be responsible for the faint EDS signals of Cr, Mn, Fe, Co and Ni, especially in the interior region of the Pt(*hkl*) substrates:

- Background signals due to the continuous X-ray component of Pt, whose intensities would depend on the sample thickness. (within a few tens of nanometres)
- Signals originating from redeposited constituent elements caused by Ga FIB processing.
- Signals induced by the material of the EDS detector. (so-called system peak mainly due to Fe and Co)

The EDS spectrum (Fig. R1a) of the marked area of the interior region of the Pt(111) substrate (yellow square region in Fig. R1b), and the quantitative values are summarised in Table R1 for the NET intensity of each constituent element. The NET intensities of the elements were 1–2 at. %, irrespective of the type kinds of elements, and the values were two orders of magnitude weaker than that of Pt. Therefore, we consider that the influence of the continuous X-ray component of Pt, redepositions, and system peaks should cause apparent EDS signals on the elemental mappings (Figs. 2b, 2d and 2f) and, thereby, processing the mapping images would no longer guarantee scientific correctness. In fact, similar phenomena have been reported in several previous studies (H. Yang *et al*, *ACS Nano*, **12**, 5 (2018) 4594-4604 and C. Yang *et al*, *Science*, **374**, 6566 (2021) 459-464). Nevertheless, the apparent faint signals located

in the interior regions of the substrates should be excluded from the discussion considering the possibility of thermal diffusions during sample preparation.

Because the discussion in the original manuscript focuses on the elemental distribution in the deposition layer rather than that in the Pt substrate, where the aforementioned factors can intervene, we have added a sentence to the revised manuscript to emphasise our view (P.8 l.20-22).

Fig. R1 a, EDS integrated spectrum for the Pt substrate region of as-synthesised Pt/Cr-Mn-Fe-Co-Ni/Pt(111). The colour-coded lines represent the energies of the characteristic X-rays of Pt $L\alpha$ (2.05 keV), Cr $K\alpha$ (5.41 keV), Mn $K\alpha$ (5.89 keV), Fe $K\alpha$ (6.40 keV), Co $K\alpha$ (6.92 keV) and Ni $K\alpha$ (7.47 keV). **b**, The EDS integrated spectrum (a) corresponds to the area marked by the yellow square in the cross-sectional HAADF-STEM image.

Table R1 NET intensity and relative quantitative ratios of each element calculated from the integrated spectrum in Fig. R1a.

Element / Series	NET Intensity (CPS)	C (wt.%)	C (at.%)
Cr K α	434	0.31	1.12
Mn K α	364	0.28	0.95
Fe K α	890	0.71	2.34
Co K α	732	0.63	1.97
Ni K α	249	0.22	0.69
Pt L α	64582	97.86	92.94

4. Figure 3: All synthesized catalysts have an increased Hads/des region after cycling indicating significant changes in the alloy surface. How did the authors normalize the currents? how did they determine the electrochemical surface area? Additionally, the authors must show actual ORR polarization curves, it is quite unusual not to display them prominently in an electrocatalysis manuscript. Have the potentials been ORR corrected? How was the uncompensated resistance determined? Please show the EIS spectra. All these factors are crucial for understanding HEA activity, as it has been outlined (e.g. *ChemEngineering* 2022, 6, 19; *ACS Catal.* 2021, 11, 1014–1023; *Appl. Surf. Sci.* 2020, 533, 147471).

Reply: The electrochemical active surface area of the UHV-synthesised Pt/Cr-Mn-Fe-Co-Ni/Pt(*hkl*) was defined using a Karletz-made O-ring (0.0903 cm²), and current values were divided by the geometric area (0.0903 cm²) to calculate the current densities for the CVs (Figs. 3a, 3c, 3e, S6a, S6c and S6e) and ORR activities (Figs. 3b, 3d, 3f, S6b, S6d and S6f). Therefore, as stated by the reviewer, iR-corrected (correction method is described below) LSV curves have been added to Figs. S7a, S7c and S7e in the Supplementary Information. The diffusion-limited current density at 0.4 V for all the corrected LSVs are within 5.9±0.1 mA/cm², suggesting that the geometrical surface area remained nearly unchanged during the measurements.

It is hardly possible to correctly evaluate the active surface areas on Pt-based alloy single-crystal surfaces, for example, Pt-HEA model catalysts, using electrochemical methods. Generally, the most common evaluation method for the electrochemical active surface area (ECSA) of pure Pt is based on hydrogen adsorption (H ads) charges per unit area for each single crystal face (for example, (111): $241 \mu\text{C}/\text{cm}^2$). However, PC-loading of the alloy surface results in dealloying of the elements, which is accompanied by the introduction of steps on the topmost surfaces and the generation of island-like structures, as shown in Figs. 4a, 4d and S5c. Such atomic and/or nano-level surface structural changes, of course, influence the H ads behaviour. The carbon monoxide (CO) stripping method is another possible electrochemical approach to estimate ECSA (H. Schmies *et al.*, *Adv. Energy Mater.*, **8**(4) (2018) 1701663 1-13 and S. Rudi *et al.*, *Electrocatalysis*, **5** (2014) 408-418). However, once CO molecules are introduced into the solution, the solute CO adsorbs on the Pt surface (surface poisoning), and thus the ORR activity trends during the PC-loading cannot be evaluated. Therefore, we discuss the ORR activity trends based on the geometrical surface area (0.0903 cm^2).

As suggested by the reviewer, we collected an EIS spectrum of as-synthesised Pt/Cr-Mn-Fe-Co-Ni/Pt(111) using our electrochemical set-up to estimate the uncompensated resistance of the LSV measurements for the RDE setup (without disk rotation (0 rpm)); the results are summarised in Fig. R2. The electrode internal resistance ($< 0.1 \Omega$), as compared to the solution resistance (82.6Ω), was negligible and was too small to be displayed in the EIS spectrum (R. Singh *et al.*, *J. Electrochem. Soc.*, **162**, 6 (2015) F489-F498 and X. Zheng *et al.*, *RSC Adv.*, **6**, 69 (2016) 64155-64164). Notably, ORR activity trends for the iR-collected ORR activity trends for Pt/Cr-Mn-Fe-Co-Ni/Pt(*hkl*) and Pt/Co/Pt(*hkl*) did not affect our conclusion (higher ORR properties of Pt/Cr-Mn-Fe-Co-Ni/Pt(*hkl*) than Pt/Co/Pt(*hkl*)). Taking into account for the aforementioned EIS results, we have added some sentences in the discussion section of

the revised main and Supplementary Information manuscripts. (Main manuscript: P.13 l.8-10; P.14 l.6, l.11, l.13-16; P.16 l.10-11; Supplementary Information: P.6 l.9-10; P.13 l.1-P. 14 l.6; P.13 Figs. S7)

Fig. R2 The Nyquist impedance diagrams of as-synthesised Pt/Cr-Mn-Fe-Co-Ni/Pt(111); the EI spectrum was recorded in an O₂-saturated 0.1 M HClO₄ solution without disk rotation (0 rpm). The frequency range was 1–1,00,000 Hz and the potential of the working electrode was fixed at 0.90 V vs. RHE.

5. The authors repeatedly use the term "PC-loading" and "potential cycles loading". Is this the same as regular cyclic voltammetry? As far as I understand yes - then please use standard terminology. If not, please elaborate the difference.

Reply: The potential cycle used in this experiment (newly added protocol schematic illustration in *Fig. 3b inset*, and revised explanation in *P.23 l.19-21*) is based on the accelerated degradation test protocol for fuel cell cathode catalyst evaluation using a half-cell geometry established by the Fuel Cell Commercialization Conference of Japan (FCCJ) (https://www.nedo.go.jp/library/PEFC_CELL_Protocol.html, R-2 III-2-2). We adopted 0.6 V (3 s)–1.0 V (3 s) square-wave potential cycles in this study. One might notice that, to the best of our knowledge, the potential loading cycles used in the degradation tests varied according to each article, depending on the country, region and affiliation of the authors (Please see for example, H. Schmies *et al.*, *Adv. Energy Mater.*, **8**, 1701663 (2018) 1-13; S. Li *et al.*, *J. Catal.*, **383** (2020) 164-171; S. Polani *et al.*, *ACS Catal.*, **11** (2021) 11407-11415 and J. Lim *et al.*, *J. Mater. Chem. A*, **10**, 13 (2022) 7399-7408).

6. Page 16: *The authors claim that their system is favorable for practical applications compared to the Pt-Co system. Considering the lack of stability of the presented catalysts I consider this claim needs to be more substantially justified, or removed.*

Reply: Notably, the atomic-level structures of the topmost surface (atomic arrangements) of single-crystal alloy surfaces easily degrade, leading to a rapid decrease in activity. Considering the above, the material properties (catalytic properties) exhibited by the Pt/Cr-Mn-Fe-Co-Ni/Pt(*hkl*) single-crystal surfaces are superior to those of Pt/Co/Pt(*hkl*) benchmark surfaces synthesised using the same fabrication procedure. Therefore, Pt/Cr-Mn-Fe-Co-Ni HEA system is surely promising material as high-performance ORR catalysts. We added some sentence to the main manuscript to reinforce our conclusion (*P.15 l.9-P.16 l.6; P.17 l.2-5*). However, the findings obtained in this study only describe the material potential of the

Pt-HEA system as a novel ORR catalyst and according to the reviewer's suggestion, the wording in the original manuscript has been modified (P.18 l.14-20).

7. Page 18: the authors claim that the catalysts display a core-shell structure, however the structure they observe is not what is classically called a core-shell structure (Nat. Chem. 2010, 2, 454–460; Nat. Mater. 2007, 6, 241–247; Rev. Chem. Eng. 2009, 25; J. Electroanal. Chem. 2003, 554-555, 191–199; J. Am. Chem. Soc. 2006, 128, 8813–8819)

Reply: We agree with the reviewer's suggestion. The degraded model catalyst surfaces presented in Figs. 4a, 4d and 4g show that PC-loading enhanced the phase separation of the surface Pt-enriched (*hkl*) and underlaid Cr-Mn-Fe-Co-Ni-rich (*hkl*) layers. In other words, the stacking structures of the underlaid Cr-Mn-Fe-Co-Ni-rich (*hkl*) and surface Pt-enriched (*hkl*) lattices became clearer in the STEM images and EDS-line profiles, as compared to those of the as-synthesised state (Figs. 4c, 4f and 4i). Therefore, as suggested, some of the wording, such as “pseudo-core-shell structure,” in the original text has been revised and the following descriptions have been inserted where it first appeared in the manuscript. “*The structure consists of surface Pt-enriched and underlaid Cr-Mn-Fe-Co-Ni-rich layers.*” (P.18 l.9-11, l.17; P.19 l.12).

Reviewer #3 (Remarks to the Author):

In the manuscript the authors report the preparation of atomic-level controlled platinum high-entropy alloy (HEA) surfaces on substrate using arc plasma deposition method and the study of the electrocatalytic properties in oxygen reduction reaction (ORR) to demonstrate this APD method for preparing experimental study platform of HEAs. Overall, I think this is a good quality paper. HEAs have attracted considerable research interest, with many exotic properties reported but the mechanisms not well elucidated. This work provides a method to finely control HEA composition and surface exposure, which can serve as model structure in fundamental studies. I think it can be considered for acceptance after the following questions are addressed.

1. It is very interesting the HEA layer as well as Pt overlayer are epitaxially deposited, i.e., with the surface plane remaining the same as the substrate surface plane. Considering arc plasma ions carries relatively high kinetic energies, I would assume the ions would sputter the substrate during deposition process and the deposition process would be dominated more by deposition kinetics rather than thermodynamics, which thus would cause surface roughness. Would the authors elaborate more regarding the epitaxial deposition mechanism?

Reply: In the arc-plasma deposition (APD) method, the target materials (metals or alloys) are evaporated by arc-plasma in a vacuum and are accelerated by the Lorentz force, and then the materials are deposited onto the substrate. Therefore, as suggested by the reviewer, the surface of the deposited material may be roughened by the deposition of elements on the substrate surface. However, the cross-sectional HAADF-STEM images of the model catalysts

with atomic resolution exhibited layer-by-layer epitaxial growth, that is, the surface Pt(111) and underlaid Cr-Mn-Fe-Co-Ni(111) lattices were stacking on the Pt(111) substrate surface. The results indicate that the eventual deposition morphology is controlled by a mechanism similar to that of the solid-phase epitaxial growth. We tuned the deposition rates of Pt and Cr-Mn-Fe-Co-Ni, the bias voltage of the APD, and post-annealing temperatures to obtain layer-by-layer stacking structures of Pt- and Cr-Mn-Fe-Co-Ni-rich (*hkl*) lattices on the corresponding Pt(*hkl*) substrates. For example, 3.0 nm-thick Cr-Mn-Fe-Co-Ni and 1.2 nm-thick surface Pt layers were deposited at 600 s and 120 s, respectively, when a minimum bias voltage (70 V) and post-annealing temperature of 773 K for Cr-Mn-Fe-Co-Ni and 623 K for surface Pt were used to synthesise the layer-by-layer stacking samples.

Cross-sectional HAADF-STEM images of a Pt/Cr-Mn-Fe-Co-Ni/Pt(111) sample synthesised using the same deposition rates and bias voltage as shown in Fig. 2a but using 473 K as the post-annealing temperature for surface Pt were newly added to Figs. S8 in the Supplementary Information. The atomically-resolved images clearly show that the layer-by-layer stacking structure became obscure and was accompanied by stacking faults and atomic defects in the sample. The atomically-resolved STEM images for the low post-annealing temperature sample clearly indicate that the post annealing temperatures (thermodynamic conditions) determine the layer-by-layer epitaxial stacking structures on the surface Pt and the underlaid Cr-Mn-Fe-Co-Ni lattices. Therefore, we added sentences and figures to the revised manuscript (Supplementary Information: P.15 l.1-17; Fig. S8).

2. The Pt/HEA layer surface became rough after ORR stability test. The authors attributed this to significant element leaching as well as element segregation in the as-prepared Pt/HEA layer. I was wondering if HEA elements were already in the as-prepared Pt overlayer and there

was significant strain in the as-prepared Pt/HEA layer, which could also contribute to the Pt surface roughing during CV cycling in ORR test condition. Would the authors comment on these possibilities?

Reply: We believe that the three-dimensional migration of the surface Pt atoms caused by the redox of Pt (so-called place-exchange) is a direct reason for the structural degradation of the topmost surfaces of the sample induced by the PC-loading conditions. Additionally, as pointed out by the reviewer, less-noble constituent elements (in this study, Cr, Mn, Fe, Co and Ni) are indeed located in the surface Pt-enriched layers (Figs. 3b, 3d and 3f) and their dissolution during PC-loading results in severe degradation of the Pt surface accompanying the release of the local strain of Pt adjacent to the less-noble elements. STEM images of 5,000 PC-loaded Pt/Cr-Mn-Fe-Co-Ni/Pt(*hkl*) (Fig. 4) clearly show that the Z-contrast induced by the difference between heavier Pt and lighter less-noble atoms becomes clearer than in the as-synthesised state, suggesting that the surface-enriched Pt and underlaid Cr-Mn-Fe-Co-Ni-rich layer phase separation (described as “pseudo-core-shell structure”) behaviour is enhanced by PC-loading. This result might indicate that less-noble constituent elements diffused to the topmost surface and dissolve into the solution during PC-loading. This behaviour of less-noble elements might be driven by the oxygen potential at the interface of the sample and solution. In contrast, the behaviours accompanying the generation of the “pseudo-core-shell structure” may suppress the sample surface degradations, as compared to Pt/Co/Pt(*hkl*). We also demonstrated that the easier thermal diffusion of Cr and Mn than Fe, Co and Ni in the as-synthesised state might suppress leaching out of Fe, Co and Ni located underneath the surface Pt, which greatly influences the electrochemical structural stability (ORR durability) of the Pt-

Cr-Mn-Fe-Co-Ni HEA system. To emphasise this, we have added some sentences in the revised manuscript (*P.15 l.9-P.16 l.6; P.17 l.2-5*).

Reviewer comments second round

Reviewer #3 (Remarks to the Author):

The authors have appropriately addressed my review comments in the last round. I think the manuscript now has an improved quality for publication and thus recommend it for an acceptance.

Reviewer #4 (Remarks to the Author):

I have reviewed the responses to the reviewer #2 comments, and feel that, in most cases, the comments have been adequately addressed. However, comment #4 was not adequately addressed, and in my judgment, this comment is crucial in assessing the validity of the claims. As reviewer #2 correctly noted, the marked changes in the Hads/des region during potential cycling attest to apparent changes in the electrochemical surface area (ECSA). However, all kinetic analysis is based on the geometric surface area. This is a serious shortcoming, as it appears likely that the true ECSA was increased due to nanoscale roughening. Hence, the authors are overestimating the ORR kinetic activity of the surfaces after potential cycling. The proper way to measure the true ECSA would be through CO stripping. The authors commented on this possibility in their response to reviewer #2, but discarded it: "The carbon monoxide (CO) stripping method is another possible electrochemical approach to estimate ECSA (H. Schmies et al., *Adv. Energy Mater.*, 8(4) (2018) 1701663 1-13 and S. Rudi et al., *Electrocatalysis*, 5 (2014) 408-418). However, once CO molecules are introduced into the solution, the solute CO adsorbs on the Pt surface (surface poisoning), and thus the ORR activity trends during the PC-loading cannot be evaluated." This analysis is incorrect and reveals a surprising misconception by the authors. CO is commonly used to assess Pt surfaces and does not poison Pt surfaces under ORR conditions. CO is readily oxidized and removed from the surface at ORR-relevant potentials. The authors should conduct the experiment as follows: CO should be purged with the potential held at a low value such as 0.08 V, followed by N₂ purging to remove the dissolved CO. The CO should subsequently be oxidized in an LSV scan to around 1 V. The background-subtracted charge associated with the CO oxidation should be used to calculate the ECSA. Following this LSV, all CO will have been removed from the surface, and the ORR kinetic can subsequently be measured without any influence of CO. I recommend major revision of the article to add CO stripping ECSA measurements for all the samples, and calculation of ORR kinetics based on the CO stripping ECSA, in place of the geometric surface area

Replies to Reviewers # NCOMMS-22-38382A

Reviewer#4

Thank you for your time and effort in reviewing our submitted manuscript. We appreciate your constructive suggestions and comments. We have carefully considered the comments and revised the main manuscript and Supplementary Information, accordingly. We have also prepared point-by-point responses to each comment below. Some of the wording has been modified.

Note: The revised parts are highlighted (yellow) in the **marked-up** main manuscript and Supplementary Information. Your comments are presented in blue in the following text. Additionally, we provide the locations for the changes in the manuscripts as *P.* and *l.*, for example, *P. 1 l. 6-12*, which refers to page and line numbers.

Reviewer #4 (Remarks to the Author):

Comments: I have reviewed the responses to reviewer #2 comments and feel that, in most cases, the comments have been adequately addressed. However, comment #4 was not adequately addressed, and in my judgement, this comment is crucial in assessing the validity of the claims. As reviewer #2 correctly noted, the marked changes in the Hads/des region during potential cycling attest to apparent changes in the electrochemical surface area (ECSA). However, all kinetic analysis is based on the geometric surface area. This is a serious shortcoming, as it appears likely that the true ECSA was increased due to nanoscale roughening. Hence, the authors are overestimating the ORR kinetic activity of the surfaces after potential cycling. The proper way to measure the true ECSA would be through CO stripping. The authors commented on this possibility in their response to reviewer #2, but discarded it: "The carbon monoxide (CO) stripping method is another possible electrochemical

approach to estimate ECSA (H. Schmies et al., Adv. Energy Mater., 8(4) (2018) 1701663 1-13 and S. Rudi et al., Electrocatalysis, 5 (2014) 408-418). However, once CO molecules are introduced into the solution, the solute CO adsorbs on the Pt surface (surface poisoning), and thus the ORR activity trends during the PC-loading cannot be evaluated.” This analysis is incorrect and reveals a surprising misconception by the authors. CO is commonly used to assess Pt surfaces and does not poison Pt surfaces under ORR conditions. CO is readily oxidised and removed from the surface at ORR-relevant potentials. The authors should conduct the experiment as follows: CO should be purged with the potential held at a low value such as 0.08 V, followed by N₂ purging to remove the dissolved CO. The CO should subsequently be oxidised in an LSV scan to around 1 V. The background-subtracted charge associated with the CO oxidation should be used to calculate the ECSA. Following this LSV, all CO will have been removed from the surface, and the ORR kinetic can subsequently be measured without any influence of CO. I recommend a major revision of the article to add CO stripping ECSA measurements for all the samples and a calculation of ORR kinetics based on the CO stripping ECSA in place of the geometric surface area.

Reply: To further understand the ORR degradation of Pt/Cr-Mn-Fe-Co-Ni surfaces by applying PC-loading, particularly the increase in the surface area of the surface Pt layer, we conducted the suggested CO stripping voltammetry for all as-synthesised (pristine) and degraded (5,000 PC-loaded) Pt/Cr-Mn-Fe-Co-Ni/Pt(*hkl*) and Pt/Co/Pt(*hkl*) sample surfaces. The voltammetry results are presented in the revised Supplementary Information (Fig. S8). As shown, the onset potentials of CO-oxidation stripping features are around 0.4 V vs. RHE on the positive-going sweeps of the first CV cycle (blue lines), followed by the main peaks appearing between 0.6 ~and 0.8 V. Such the current responses (peaks) were located on the negative potential side compared to the clean Pt surface [P. Inkaew et al., *J. Electrochem. Soc.*, **614** (2008) 93-100; N. Lebedeva et al.,

J. Electrochem. Soc., **487**, 1 (2000) 37-44.; G. Garcia and M. T. M. Koper, *Phys. Chem. Phys.*, **10** (2008) 3802-3811; and E. G. Ciapina *et al.*, *J. Electrochem. Soc.*, **815** (2018) 47-60]. Therefore, the negative shifts of the peaks stem from the underlaid alloying metal elements. [P. Ochal *et al.*, *J. Electroanal. Chem.*, **655** (2011), 140-146] Notably, the CO-oxidation stripping charges (1st cycle of CO oxidation response subtracted by the 2nd CV cycle) were almost the same for all the as-synthesised pristine sample surfaces of Pt/Cr-Mn-Fe-Co-Ni/Pt(*hkl*) and Pt/Co/Pt(*hkl*). The voltammetry results clearly indicate that the effective ECSAs of the as-synthesised samples were almost equivalent to the geometrical surface area confined by the O-ring (0.0903 cm²; as described in the Methods of electrochemical (ORR) property evaluation in the main manuscript), and that the vacuum-synthesised Pt/Cr-Mn-Fe-Co-Ni/Pt(*hkl*) and Pt/Co/Pt(*hkl*) surfaces were almost atomically flat at the pristine state. Therefore, increases in the effective ECSA by applying PC-loading (corresponding to degraded states after 5,000 PCs) indicate that the actual surface area increases through surface roughening events induced by PC-loading. As shown in the bar graphs of Fig. S8b, the CO-oxidation stripping charges for all sample surfaces increased with 5,000 PC-loading, irrespective of the surface orientations and of underlaid alloying elements. Indeed, increases in the effective ECSA at each degraded state (estimated after 5,000 PCs) correspond well with the atomic-level surface roughness induced by the same PC-loading, as shown in the corresponding cross-sectional HAADF-STEM images (Figs. 4 and S5). Meanwhile, ORR deactivation was caused by applying PC-loading. i.e., decrease in the specific ORR activities calculated based on the geometrical surface area (green) or on effective ECSA (red in Fig. S8c) are qualitatively similar. Therefore, even considering the effective ECSAs estimated by CO-stripping voltammetry, the ORR deactivation behaviours obtained by applying PC-loading are surely consistent with the results shown in the main manuscript (Fig. 3) for the ORR activity trends discussed based on the geometrical surface area. In particular, the estimated effective ECSAs for Pt/Cr-Mn-Fe-Co-

Ni/Pt(111) in the degraded state (after 5,000 PCs) were clearly smaller than those of Pt/Co/Pt(111) (Fig. S8b), clearly indicating the suppression of the atomic-level surface structural degradation by PC-loading. This also emphasises the superior ORR properties of the Pt/Cr-Mn-Fe-Co-Ni high entropy alloy surface.

We have added some sentences and datasets to the discussion section of the revised main manuscript and Supplementary Information. (Main manuscript: *P.15 l.19-P.16 l.9; P.17 l.4-8; P.23 l.19-P.24 l.3*; Supplementary Information: *P.15 l.1- P.17 l.6, P.15 Figs. S8*)

Reviewer comments third round

Reviewer #5 (Remarks to the Author):

The author have appropriately addressed concerns of Reviewer #4, and the added CO stripping data improves the quality of this paper.